# Recent Insights into Sample Pretreatment Methods for Mycotoxins in Different Food Matrices: A Critical Review on Novel Materials

**DOI:** 10.3390/toxins15030215

**Published:** 2023-03-10

**Authors:** Yu Bian, Yuan Zhang, Yu Zhou, Binbin Wei, Xuesong Feng

**Affiliations:** 1School of Pharmacy, China Medical University, Shenyang 110122, China; 2School of Traditional Chinese Materia Medica, Shenyang Pharmaceutical University, Shenyang 110016, China; 3Department of Pharmacy, National Cancer Center/National Clinical Research Center for Cancer/Cancer Hospital, Chinese Academy of Medical Sciences and Peking Union Medical College, Beijing 100021, China

**Keywords:** mycotoxins, sample pretreatment, microextraction techniques, novel materials, food

## Abstract

Mycotoxins pollution is a global concern, and can pose a serious threat to human health. People and livestock eating contaminated food will encounter acute and chronic poisoning symptoms, such as carcinogenicity, acute hepatitis, and a weakened immune system. In order to prevent or reduce the exposure of human beings and livestock to mycotoxins, it is necessary to screen mycotoxins in different foods efficiently, sensitively, and selectively. Proper sample preparation is very important for the separation, purification, and enrichment of mycotoxins from complex matrices. This review provides a comprehensive summary of mycotoxins pretreatment methods since 2017, including traditionally used methods, solid-phase extraction (SPE)-based methods, liquid-liquid extraction (LLE)-based methods, matrix solid phase dispersion (MSPD), QuEChERS, and so on. The novel materials and cutting-edge technologies are systematically and comprehensively summarized. Moreover, we discuss and compare the pros and cons of different pretreatment methods and suggest a prospect.

## 1. Introduction

Mycotoxin is a low-molecular-weight (about 700 Da) secondary metabolite produced when toxic fungi infect crops, which can cause contamination of many food and feed products [1]. Generally, crops stored for more than a few days will become potential targets for mycotoxin formation. The main crops affected are cereals, nuts, dried fruits, coffee, spices, oilseeds, beans, and fruits, especially apples [2]. According to a Food and Agriculture Organization (FAO) report, 25% of grains were contaminated by mycotoxins [3]. In addition, mycotoxins may be found in beer and wine because of the use of contaminated barley or grapes in the production process [4]. Because livestock eat contaminated feed, they can also enter the human food chain through meat or other animal products like eggs and milk [4].

The major mycotoxins are aflatoxins (AFs), citrinin (CIT), fumonisins (FBs), trichothecenes (TCTs), ochratoxins (OTs), patulin (PAT), and zearalenone (ZEA). Based on the structural diversity (Appendix A), mycotoxins show a variety of toxicological characteristics, and are the chief culprit of many diseases. The main classification of mycotoxins, representative toxins of each type, and their toxicological effects are shown in Appendix A [2,5,6,7,8,9,10]. Compared with these strictly regulated mycotoxins, the emerging mycotoxins including *Fusarium* mycotoxins (such as beauvericin (BEA), enniatin (ENNs), moniliformin (MON), fusaric acid (FA), Alternaria mycotoxins (alternariol (AOH), alternariol methyl ether (AME)), tenuazonic acid (TeA), tentoxin (TEN), and altenuene (ALT)) have not been strictly regulated, but they can also cause serious harm to the human body, so emerging mycotoxins are also a public health problem [11]. Therefore, it is very necessary to develop a simple, rapid, green, accurate, and high-throughput mycotoxin sample pretreatment method for strict safety monitoring and maintenance of human health.

In the past five years, only four reviews concerning the pretreatment methods for mycotoxins have been updated. In 2017, Alshannaq et al. [12] summarized the occurrence, toxicity, and analysis of major mycotoxins in food and pay more attention to different determination methods. Next, Tahir et al. [13] and Zhang and Banerjee [14] summarized the extraction, analysis and control methods of AFs, as well as the sample preparation and chromatographic techniques of AFs. In 2022, Tang et al. [15] focused on the SPE techniques based on nanomaterials for mycotoxin analysis in food and agricultural products. It is worth noting that in recent years, many new materials, novel solvents, and emerging technologies have been continuously developed and applied to the sample preparation of mycotoxins, but there is no corresponding review so far.

In the current article, the literature on the pretreatment of major and emerging mycotoxins in the past five years is critically summarized, and this comprehensive update is finally completed. The purpose of this review is to: (i) provide an update for the sample pretreatment method of mycotoxins (such as SPE-based approaches, dispersive liquid-liquid microextraction (DLLME), and matrix solid phase dispersion (MSPD)); (ii) summarize the novel materials and solvents developed recently used in pretreatment methods for mycotoxins; (iii) compare the advantages and disadvantages of different methods, and put forward future development trends for better advancement; and (iv) help scientists in this field better understand and implement existing methods, and promote more progress through technical improvement.

## 2. Sample Pretreatment Methods

In the process of analyzing samples, pretreatment is necessary, time-consuming, and a weak link, which directly affects the accuracy, reproducibility, and detecting limit of detection (LOD) results. Appropriate sample pretreatment methods can (1) eliminate potential matrix interference, thus exposing trace substances and prolonging the service life of the instrument, and (2) enrich the target analyte from the complex matrix, thus meeting the needs of detection methods. With the emergence of new technologies and novel materials, sample pretreatment methods are also developing. Appendix A [11,16,17,18,19,20,21,22,23,24,25,26,27,28,29,30,31,32,33,34,35,36,37,38,39,40,41,42,43,44,45,46,47,48,49,50,51,52,53,54,55,56,57,58,59,60,61,62,63,64,65,66,67,68,69,70,71,72,73,74,75,76,77,78,79,80,81,82,83,84,85,86,87,88,89,90,91,92,93,94,95,96,97,98,99,100,101,102,103,104,105,106,107,108,109,110,111,112,113,114,115,116,117,118,119,120,121,122,123] summarizes in detail the sample pretreatment methods used for mycotoxins since 2017.

### 2.1. Traditional Used Methods

Solid-liquid extraction (SLE) is often used to extract mycotoxins from different samples because of its simple operation and because it does not need any instruments. For dry substrates, hydration is usually needed to wet and swell the sample to ensure the effective extraction and separation of mycotoxins deposited in the sample, and also contributes to the subsequent clean-up steps. The choice of solvent is closely related to the sample matrix and the target species. In order to improve the extraction efficiency of mycotoxins, it is necessary to optimize the types of solvents. In the study of Huertas-Pérez et al. [16], satisfactory results were achieved for AFs when 100% acetonitrile (ACN) was used in white rice (recoveries >85%, relative standard deviations (RSDs) < 5%), while 100% ACN as an extraction solvent yielded unacceptable recoveries (44–80%) for AFs in brown rice. To solve this problem, the ratio of extraction solvent was optimized. Finally, extraction of AFs was achieved by adding 100% can extraction solvent for white rice and ACN:H_2_O (8:2, *v*/*v*) for brown rice samples with recoveries in the range of 84.5 and 105.3% and RSD% lower than 6%. Paschoal et al. [37] developed a rapid single-extraction method for the simultaneous analysis of AFB1, B2, G1, G2, FB1, and ZEA in corn meal. Three different extraction solvents, including ACN/H_2_O (80:20, *v*/*v*), ACN/methanol (MeOH)/H_2_O (60:20:20, *v*/*v*/*v*), and MeOH/ACN/H_2_O (60:20:20, *v*/*v*/*v*) were tested. For AFB1, B2, G1, G2, and ZEA, using all three solvents results in a high recovery (74.97–104.25%). Notably, FB1 has a higher molecular weight and presents several interaction sites in its molecular chain, which enables it to interact more with the matrix, thus increasing the difficulty of extraction. Compared with ACN, MeOH is more polar and acidic, and the recovery is higher when a higher proportion is used in the extraction process. Therefore, MeOH/ACN/H_2_O (60:20:20, *v*/*v*/*v*) was chosen as the extraction solvent, and good recoveries (75.16–97.66%) were obtained.

In order to improve the extraction efficiency and shorten the extraction time, ultrasonic assisted extraction (UAE) is often used to extract mycotoxins [52,57]. Sartori et al. [52] established a simple UAE method to extract 14 mycotoxins from cereal-based porridge destined for infant consumption. After rapid shaking and sonication (10 min), without any further clean-up steps, recoveries ranging from 63.5 to 113.2% were obtained.

However, traditional organic solvents such as ACN and MeOH are harmful to the human body and the environment. With the development of green chemistry, deep eutectic solvent (DES) as a new type of green solvent and a substitute for traditional organic solvents has attracted extensive attention. DES is obtained by the interaction of two or more safe, low-cost, and biodegradable components through hydrogen bonding. They have good characteristics such as non-toxicity, environmental friendliness, simple preparation, high thermal stability, and low vapor pressure and design ability, and is considered as an environmentally friendly and efficient extractant [124]. He et al. [17] developed a simple strategy based on DES with tetramethylammonium chloride-malonic acid (n/n, 1:2) for extraction of AFs in rice samples. After optimization, good precision (RSD < 5.52%) and recoveries (78.93–113.64%) were obtained, which indicated that DES could be applied as an efficient extractant for AFs analysis in rice samples.

Pressurized liquid extraction (PLE) uses high temperature and high pressure to maintain the solvent state in the system. The increase of pressure is helpful for improving the solvation and extraction kinetics, and the increase of temperature will destroy the interaction between targets and matrixes, thus improving extraction efficiency. As a subclass of PLE, subcritical water extraction (SWE), also known as pressurized hot water extraction (PHWE), uses pure water as solvent, which is an environmentally friendly separation method. Another superiority of using water as a solvent is that it allows subsequent selective washing of the resulting extract without any prior exchange of solvents, thus shortening the analysis time. In this respect, including the cleaning step can reduce or even prevent the matrix effect (ME). Miró-Abella et al. [32] put forward a SWE method to extract six of the most abundant trichothecenes in cereals, a pseudocereal, and an oilseed. Water containing 1% formic acid was used as extraction solvent, and then SPE was used for cleaning without any solvent exchange, which achieved good extraction performance and acceptable extraction recovery.

### 2.2. SPE-Based Approaches

#### 2.2.1. Solid-Phase Extraction (SPE)

##### Traditional Materials Used in SPE

SPE is mainly used for the separation, purification and enrichment of targets, with the main purpose of reducing the interference of the sample matrix and improving the detection sensitivity. Currently, it is one of the most popular sample pretreatment methods in food, environment, and other matrices. Compared with traditional methods, SPE has the advantages of low solvent consumption, effective preconcentration, and high recovery. In addition, various new stationary phases, microextraction technologies and on-line technologies have given SPE more room for development. The traditional SPE process includes four steps: (1) precondition, (2) sample loading, (3) washing, and (4) eluting. Figure 1A shows the common procedure of SPE. In addition, we summarize recent assays that use SPE and the main analytical parameters in Table 1.

Adsorbent is the core of the SPE method. According to the polarity of materials, commercial SPE adsorbents can be divided into four categories: (1) reversed-phase adsorbents for nonpolar or weakly polar compounds, such as C_18_ and C_8_, (2) positive-phase adsorbents for polar compounds, such as CN, NH_2_, and Florisil, (3) ion-exchange adsorbents, such as SAX, PRS, and PSA, and (4) additional adsorbents, such as Oasis HLB, immunoaffinity columns (IACs), and PRiME [125]. As shown in Table 1, for the extraction of mycotoxins, commercial HLB [67,85], SAX [121] and IACs [69,96,104,115] are the most commonly used adsorbents. In the process of clean-up, it is usually necessary to optimize the adsorbent to obtain better results. Dong et al. [85] compared the purification effects of 2 SPE cartridges and 3 sorbents in QuEChERS clean-up of 7 mycotoxins in fruits and vegetables. The results showed that the QuEChERS method cannot effectively remove pigments, and the data of recovery was very poor whether PSA, C_18_, or GCB was used as the adsorbent. Samples cleaned up with C_18_ SPE cartridges showed poor recoveries (0–68%), which was attributed to the compounds’ loss during the purification process. By contrast, the HLB SPE cartridge obtained highest recoveries (85–108%), so it is selected for purification.

IAC is based on the adsorption mechanism of the interaction between antigen and antibody, which shows high specificity. SPE method with IAC as adsorbent has been widely used in mycotoxin analysis. Gab-Allah et al. [30] used T2/HT2 IAC as SPE adsorbents to clean T-2 and HT-2 toxins in cereal products. The sorbent was rinsed using pure water, and then ethanol was applied onto the IAC to elute the targets. The results showed good recoveries of T2 and HT2 in the range of 78.6–98.6%, while RSDs for intraday and interday precisions were below 7%. Furthermore, excellent method selectivity was obtained since there were no matrix interference peaks over the entire chromatographic run. In addition, the IAC column can be combined with other columns to further improve the purification and enrichment effect. Ye et al. [108] used a multifunctional purification column (MFC) and IAC in tandem to clean AFB1, B2, G1, and G2 in dark tea. Using the MFC-IAC cleaning method, the matrix interference was effectively reduced. Recoveries ranged from 77.5 to 93% with RSDs <11.0%.

##### From Traditional to Novel Materials in SPE Columns

Molecular recognition-based SPE columns. Although IAC usually has high selectivity and less background interference, the wide application of IAC is limited due to the lack of antibodies, such as sensitivity to the sample matrix, cross-reactivity of antibodies, high cost, poor tolerance to organic solvents, and the difficulty in obtaining them. Therefore, the development of new SPE adsorbents is a hot research topic at present. As another selective adsorbent, molecularly imprinted polymers (MIPs) are considered to be very promising, and can replace IAC because of their higher stability and capacity, as well as their lower cost. The identification process of MIPs is based on the “key and lock principle”. After polymerization and template extraction, a three-dimensional structure cavity is formed. Because the cavity is complementary to the targets in size, shape, and chemical function, MIPs have good recognition ability to these targets. In addition, MIPs have the advantages of easy preparation, high mechanical, thermal, and chemical stability and resistance to a wide range of pH values and temperatures. Because of these special characteristics, MIPs are an ideal SPE adsorbent for sample cleaning, namely MISPE. Lhotská et al. [105] developed a MISPE method with commercial ZEA-selective SPE columns to purify ZEA in beer samples. After optimization, satisfactory recoveries (99.0 ± 2.5%) were achieved.

In addition, unlike commercial SPE cartridges, the MIP adsorbents could be reused many times without losing their recognition properties. In the work of Rico-Yuste et al. [60], MISPE was used to analyze AOH and AME in foodstuffs. 4-vinylpyridine (VIPY) and methacrylamide (MAM) were used as functional monomers, ethylene glycol dimethacrylate (EDMA) was chosen as a crosslinking agent, and 3,8,9-trihydroxy-6H-dibenzo[b,d]pyran-6-one (S2) was employed as an AOH substitution template. Recoveries ranging from 92.5 to 106.2% were obtained in tomato juice and sesame oil. After testing, the MISPE can be reused more than 30 times with satisfactory recognition properties. Moreover, compared with non-imprinted polymers (NIPs), MISPE had a stronger affinity for AOH and AME.

An aptamer is a short oligonucleotide sequence or a short polypeptide obtained by in vitro screening, which can bind with the corresponding ligand with high affinity and strong specificity. When an aptamer interacts with a target, its internal chain base pairing can form various three-dimensional (3D) structures, which make an aptamer bind to a specific region of a target. Recently, based on the specific recognition ability of “chemical antibody”, a new type of SPE adsorbent was developed by using aptamer modified carrier surface as functional monomer, which has advantages of specificity, accuracy, and repeatability for enriching ultra-trace targets. Zhang et al. [70] prepared and applied an M1-aptamer-functionalized sorbent for the adsorption of ultra-trace AFM1 and analogues in milk samples, as shown in Figure 2A. During the extraction process, the M1-aptamer on the surface of the carboxyl microspheres were completely modified via N-(3-dimethylaminopropyl)-N’-ethylcarbodiimide hydrochloride (EDC)/NHs amide formation. The adsorption capacity of this sorbent was 233.1 μg/g. Compared to carboxyl microspheres (recovery of 60.53–65.96%) and synthetic transition states (recovery of 67.98–73.11%), M1-aptamermodified microspheres showed excellent spiked recovery (86.76–103.3%).

Nanoparticles-based SPE adsorbents. Graphene is one of the most exciting two-dimensional (2D) carbon nanomaterials in analytical chemistry. It is a single layer of carbon packed in a hexagonal lattice, and the carbon-carbon distance is 0.142 nm [126] with a carbon-carbon distance of 0.142 nm. Because of its large specific surface area (theoretical value = 2630 m^2^/g) with a high pre-concentration factor, special thermal and mechanical properties, good chemical stability, and excellent conductivity, it has become an excellent adsorbent [127]. Feizy et al. [20] used graphene nanoparticles as an SPE adsorbent for the extraction and pre-concentration of the AFs in rice and wheat samples. After optimizing the parameters, 2.5 mL of ACN/H_2_O/MeOH/acetic acid (HOAC) (59.4:9.9:29.7:1 *v*/*v*) was chosen as the eluting solvent, using 20 mg of the graphene nanoparticle as the adsorbent, and 40 mM of NaCl was added to the solution. At the spiked level of 2 and 5 ng/g, the relative recoveries values of 75.88–113.30% and 70.61–110.75% were obtained in cereal samples.

However, the application of graphene is still limited because its irreversible van der Waals aggregation and π-π stacking interaction in solution lead to non-specificity, poor repeatability, and poor reliability [128]. In order to solve this problem, a second component as nano-spacer and conductor can be introduced to increase the distance between graphene layers and minimize agglomeration. Jiang et al. [71] proposed a reliable SPE procedure using reduced graphene oxide and gold nanoparticle (rGO/Au) composites as an adsorbent for purification and enrichment of 9 mycotoxins in milk. Adequate recovery between 70.2 and 111.2%, and acceptable precision with RSD ranging from 2.0 to 14.9% were obtained.

Activated carbon (AC) has the potential to adsorb a large number of compounds because of its huge surface area. However, its selectivity to the target is relatively poor. In order to improve the selectivity, the chemical properties of AC surface are usually modified. Due to the intrinsic and extrinsic properties of boron, doping boron into carbon compounds has a good influence on materials. Importantly, compared with most commonly reported metal doping, using boron to improve the efficiency of AC doping adsorbents is low-cost and non-toxic. Compared with the traditional IAC-SPE method, the activated carbon-boron (AC-B)-SPE has the advantages of environmental protection, simplicity, rapidity, sensitivity, reliability, strong selectivity, good repeatability, and easy application. Karapınar et al. [81] prepared an AC-B nanocomposite-based SPE method to extract and specify AFG1, G2, B1, and B2 in nuts. After optimizing the extraction parameters (adsorbent amount (5 mg), vortex time (3 min), pH (5), desorption solvent (ACN), desorption volume (3 mL), and desorption time (3 min)), recoveries ranging from 89.5 to 96.5% of AFs with RSDs less than 7% were obtained.

Bonded silica-based SPE adsorbents. Humic acid (HA) is a natural organic mixture produced by complex microbial degradation of plant and animal remains. The structural formula of HA contains a hydrophobic framework and rich hydrophilic groups, which enables HA to interact with some metal ions, oxides, and toxic organic compounds through different ways of interaction. Aiming at mycotoxins in oil samples, the oil matrix can be selectively washed from HA under hydrophobic conditions, and AFs can be captured by HA, thus achieving the purpose of selectively extracting AFs from oil. In addition, HA exists widely in nature, which makes the cost of the HA-bound silica (HAS) SPE column about 10% of the IAC column and 20% of the MFC column. Zhou et al. [63] used a low-cost HAS as an effective SPE sorbent for convenient pretreatment of AFs in edible oils. In the HAS-SPE program, AFs can be captured by the adsorbent through hydrophobic and hydrophilic interaction, while the oil matrix can only be captured by hydrophobic interaction. Isopropanol was used to wash off oil, thus achieving selective extraction of AFs in oil matrices. After optimization, high recoveries (82–106%) and minor MEs (99–105%) for AFB1, B2, G1, and G2 were achieved in various oil matrices.

In recent years, more and more attention has been paid to green chemistry, and low- or non-toxic sorbents are still needed urgently at present. Ionic liquids (ILs) are widely used because of their good thermal stability, miscibility with water or organic solvents, high extraction efficiency for some compounds, safety and environmental protection, and π-π interactions between analyte and functional groups of the ILs. Fang et al. [36] developed a SPE method using new bifunctional IL-based silicas as sorbents to isolate AFB1 from moldy corn and peanuts. Compared with imidazolium chloride-butylimidazolium chloride-based silica (Sil@BIm-Im), imidazolium chloride-hexylimidazolium chloride-based silica (Sil@HIm-Im) had the highest adsorption efficiency, and was used as SPE sorbent. Under the optimized conditions, 0.009 μg/g and 0.023 μg/g of AFB1 were detected in corn and peanut extracts, and recoveries of 80.0–103.3% and RSDs of 2.37–6.58% were obtained.

Hypercrosslinked polymers (HCPs). Hypercrosslinked polymers (HCPs) have a wide application prospect in many fields, such as catalysis, gas storage, and pollutant adsorption, because of their high specific surface area, mild synthesis conditions, wide sources of monomers, and cheap and easily available catalysts. It is worth noting that heterocyclic HCP containing heteroatoms can produce higher affinity. In the study of Xu et al. [18], heterocyclic phenyl-imidazole (2-phenylimidazole (PI), 4,5-diphenylimidazole (DPI), and 2,4,5-triphenylimidazole (TPI)) were used as monomer to fabricate three novel hyper-crosslinked polymers (PI-HCP, DPI-HCP and TPI-HCP) to pretreat Afs. The preparation of N-heterocyclic HCPs and their application for extraction of Afs was shown in Figure 2B. All three N-HCPs showed large surface area, good stability, and high affinity for AFs. The adsorption mechanism between N-HCP and AFs can be attributed to hydrogen bonding, polar interaction, and π-π stacking. Taking one of the materials (PI-HCP) as an example, a satisfactory recovery rate (82.2–111.0%) was obtained.

##### A new Perspective of SPE: Online Technology

To avoid time-consuming and manual handling of samples of off-line methods, it is necessary to develop new automatic programs. The new method should not only be fast, but also accurate enough to minimize errors and obtain repeatable responses. On-line SPE is a suitable alternative technology for sample preparation, because the sample pretreatment procedure is minimal, which reduces the errors of experiment and the consumption of organic reagents, shortening the analysis time. Campone et al. [105] developed a fast and automated on-line SPE-HPLC-MS/MS procedure for the pre-concentration, clean-up, and sensitive determination of OTA in wine samples. Compared with other traditional methods requiring several hours for extraction and purification, this method achieved accurate determination of OTA in less than 20 min. Satisfactory recovery (80–88%) and RSD (2–6%) were achieved. In addition, on-line SPE technology has the advantages of high sample throughput and low analysis cost.

#### 2.2.2. Magnetic Solid-Phase Extraction (MSPE)

MSPE disperses the magnetic adsorbent evenly in the sample solution, and realizes the rapid separation and enrichment of the target analyte from the sample solution with the help of external magnetic field. The MSPE method is simple to operate, and is especially suitable for the analysis of large-volume samples. Compared with SPE, MSPE effectively avoids time-consuming and tedious on-column SPE procedures, such as adsorbent packaging and sample loading, and avoids the problems of cartridge clogging. In recent years, MSPE has become an attractive sample preparation technology and attracted more and more attention. Among them, magnetic adsorbent is the most important part in MSPE, which directly determines the extraction efficiency and time. Therefore, more and more researchers focus on the synthesis of novel and efficient magnetic adsorbents, such as magnetic covalent organic frameworks (MCOFs) (Fe_3_O_4_/COF-TpBD [41], Fe_3_O_4_@COF(TFPB-PPD) [23]), magnetic metal organic frameworks (MOFs) (MIL-101(Cr)@Fe_3_O_4_ [45], Fe_3_O_4_@UiO-66-NH_2_@MON [24]), magnetic carbon nanotubes (M-CNTs) (PDA@Fe_3_O_4_-MWCNTs [64], MWCNT-MNPs [42]), magnetic hyper-crosslinked polymers (MHCP) ((MHCP-TPE) [21], M-OP10-DCX [22]), and others (Fe_3_O_4_@SiO_2_@TiO_2_-APTMS-CPA [82], Fe_3_O_4_@POSS@PIL-PSt [33]). The main analytical parameters of MSPE on the pretreatment of mycotoxins is shown in Table 2. Generally, magnetite (Fe_3_O_4_) is the most common compound to endow the obtained composite with magnetic properties.

Magnetic Covalent Organic Frameworks (MCOFs). COFs are a new kind of porous organic framework crystal material, which is mainly composed of light elements such as C, N, O, and B through covalent bonds. It is a highly crystalline material, and has attracted much attention because of its outstanding advantages such as large specific surface area, thermochemical stability, controllable chemical and physical properties, low framework density, and permanent open pore structure [129]. MCOFs can further improve the dispersibility of adsorbents, which is beneficial to its effective contact with the analyte. Moreover, the introduction of an external magnetic field overcomes the disadvantage of difficult separation from matrix solutions, saves time, and reduces the consumption of organic reagents. Li et al. [23] prepared MCOFs using 1,2,4,5-Tetrakis-(4-formylphenyl)benzene (TFPB) and p-Phenylenediamine (PPD) as monomers to purify AFB1, B2, G1, and G2 in milk, edible oil, and rice samples (Figure 2C). Under the optimized conditions, only 5 min of extraction time and 2 mL organic reagent were needed, and the recoveries ranged from 76.4 to 112.5%. In addition, the adsorbents can be reutilized more than 8 times.

In order to further shorten the extraction time and improve the extraction efficiency, MSPE was combined with auxiliary technologies. In the study of Wei et al. [41], a simple and rapid vortex-assisted MSPE method using Fe_3_O_4_/COF-TpBD as adsorbents was established to extract 10 mycotoxins in maize. Vortex was helpful to increase the contact area between magnetic adsorbent and analyte, thus accelerating the adsorption of analyte. Both the adsorption time and the desorption time only took 30 s. Satisfactory recoveries (73.8–105.3%) with RSDs less than 8.5% were obtained.

Magnetic Metal Organic Frameworks (MMOFs). MOFs are crystalline porous materials with periodic network structure formed by self-assembly of transition metal ions and organic ligands. It has advantages of high porosity, low density, large specific surface area, regular pores, adjustable pore size, and diversity of topological structure and tailoring [130]. However, traditional MOFs have cumbersome separation processes in practical applications and competitive adsorption is also a major challenge for MOFs. In order to solve the above problems, a new magnetic micro-composite Fe_3_O_4_@UiO-66-NH_2_@MON was constructed by Li et al. [24] (Figure 2D). Among them, UiO-66 is one of the most stable MOF materials, which is rich in amino groups and can be further covalently combined with other materials. As a hydrophobic “shield”, microporous organic networks (MON) can improve the moisture resistance and stability of adsorbents. Using this new adsorbent to extract AFs, satisfactory recoveries of 88.4–94.4% for corn samples, 87.3–96.7% for rice samples, and 89.5–101.8% for millet samples were obtained.

MIL-101(Cr) is composed of Cr^3+^ ion and terephthalic acid. It has a mesoporous molecular sieve structure and a large number of unsaturated Cr(III) sites, which can effectively combine electron-rich functional groups [128]. Moreover, MIL-101 (Cr) is more stable under acidic and neutral conditions than some other MOFs, thus widening their applications in different areas. Guo et al. [45] established an efficient and fast MSPE method using MIL-101(Cr)@Fe_3_O_4_ nanocomposites as an adsorbent to purify and enrich 9 mycotoxins in maize, wheat, watermelon and melon samples. The adsorption time was fixed at 4 min, and elution time was chosen at 6 min. Satisfactory recoveries (83.5–108.5%) and acceptable precision (RSD, 1.6–10.4%) were achieved.

Magnetic Carbon Nanotubes (M-CNTs). CNTs are novel and interesting carbonaceous materials, which can be divided into single-walled carbon nanotubes (SWCNTs) and multi-walled carbon nanotubes (MWCNTs) according to the number of carbon atom layers in the wall of nanotubes. CNTs have great surface area and unique structure, so they have excellent adsorption capacity. In recent years, due to unique electronic, mechanical, and chemical properties, MWCNTs have become one of the most frequently used constructive nano-materials as SPE adsorbents. Han et al. [42] developed a simple and reliable MSPE procedure using MWCNTs—magnetic nanoparticles (MWCNT-MNPs) as sorbents to simultaneously purify and enrich of ZEA and its derivatives in maize. After optimization, the MEs (92.1–103.8%) were greatly minimized, the recoveries ranged from 75.8 to 104.1%, and the inter- and intra-day precision was less than 14%. Xu et al. [64] investigated the adsorption of Afs, OTA, and ochratoxin B (OTB) in vegetable oils with polydopamine-coated magnetic (PDA@Fe_3_O_4_)-MWCNTs as adsorbents of SPE. Compared with Fe_3_O_4_-MWCNTs, PDA@Fe_3_O_4_-MWCNTs had a higher extraction efficiency. Under the optimized conditions, the recovery was 70.15–89.25% with RSD less than 6.4%, which indicated that the novel adsorbents had a high affinity toward AFs, OTA and OTB.

#### 2.2.3. Solid-Phase Microextraction (SPME)

SPME is developed on the basis of SPE, which follows the trend of miniaturization in modern analytical chemistry. Compared with SPE, SPME has the advantages of simple operation, convenient carrying, low manufacturing cost, and no filler blockage. Moreover, SPME integrates sampling, extraction and concentration, and only a syringe-like device is needed to complete the whole pretreatment process. Traditionally, carboxen (CAR)/polydimethyl siloxane (PDMS), divinylbenzene (DVB)/PDMS, and DVB/CAR/PDMS are the three most commonly used SPME fiber coatings. However, their sensitivity and selectivity are relatively poor. Recently, the research focus of SPME is new fiber materials, such as graphene- and ILs-based adsorbents [55,95], and novel extraction forms, like in-pipe SPME (IT-SPME) [25]. Amde et al. [95] synthesized [HMIM][PF6]@ZnO-NRs as the adsorbents of SPME to simply, quickly, and cost-effectively pretreat AFB1, B2, G1 and G2 in food products. After optimization (10 mg of adsorbent, pH 7, vortexing for 1 min, and desorbed by sonication for 2 min in 1 mL ACN), good recovery in the range of 88.6–99.8% was acquired.

However, traditional SPME may have the following problems: the fiber is fragile, and the coating is easy to degrade in recycling use. These common problems can be solved by IT-SPME. In addition, the IT-SPME technique show additional advantages, such as automation and on-line coupling with the detection method. Generally, the mechanism of IT-SPME in extraction, purification, and enrichment is based on the equilibrium of targets between stationary phase and sample solution. Therefore, selecting a suitable adsorbent is helpful to the pretreatment of the target. Wu et al. [25] prepared a poly (methacrylic acid-co-divinyl-benzene) [poly (MAA-co-DVB)] monolithic column, and used it for IT-SPME of the selected AFB1, ZEN, and sterigmatocystin (STEH) from rice grain. High enrichment factors (EFs) for the three mycotoxins ranging from 71.9 to 98.7, recoveries in the range of 78.0–102.8%, and RSD of 2.96–4.74% were acquired.

#### 2.2.4. Dispersive Micro-Solid-Phase Extraction (D-μ-SPE)

SPE is a widely used sample preparation method. However, this method requires multiple steps, which is time-consuming, complicated, and expensive. To solve the above problems, researchers strive to develop fast, efficient and low-cost sample preparation methods, which are in line with the main development trend of modern analytical chemistry (miniaturization and simplification). D-μ-SPE only needs micro- or nano-adsorbents to adsorb target analytes, which is economical and accords with the principle of green chemistry. The D-μ-SPE program is shown in Figure 1B. Zhu et al. [46] developed a D-μ-SPE method to purify and enrich AFs in wheat and peanut samples. Three different ILs, [OMIM][PF6], [HMIM][PF6], and [BMIM][PF6] were compared, and finally, [HMIM][PF6] functionalized nanomaterials ([HMIM][PF6]-ZnO nanoflowers (NFs)) were chosen for the next experiment. Only 10 mg of adsorbents was needed, and the satisfactory recoveries were from 93.8 to 105.1%. In addition, in the process of D-μ-SPE, the MEs can be minimized or eliminated, and the analysis of trace targets can be realized. Mohebbi et al. [74] used vitamin-based MOF as the sorbents of D-μ-SPE for the extraction of AFs from commercial soy milk samples. The method showed some advantages such as negligible ME (89–107%), short extraction time (20 min), and acceptable RSDs (3.1–4.0%).

Microwave-assisted extraction (MAE) is a green extraction technology with great development potential. With the help of microwave energy, the solvent in contact with the sample can be heated to separate the required compounds from the sample matrix and enter the solvent, which is a process of strengthening heat and mass transfer on the basis of traditional extraction technology. Through microwave strengthening, the extraction speed, extraction efficiency and extraction quality are improved, and the extraction time and the amounts of organic reagents are reduced. Du et al. [58] proposed a MA-d-μ-SPE for the extraction of 6 mycotoxins in peach seed, milk powder, corn flour and beer samples. After optimizing the experimental parameters [type of dispersant (nano zirconia), concentration of dispersant (2.5 μg/mL), vortex time (2 min), extraction time (10 min), type of desorption solvent (chloroform) and pH (4.5)], acceptable reproducibility (RSD < 4.59%) and satisfactory spiked recoveries (84.27–104.96%) were obtained.

#### 2.2.5. Micro-Solid-Phase Extraction (μ-SPE)

Microextraction has been developed rapidly over the past few years to overcome some of the limitations of classical techniques using large amounts of organic reagents and complex operating procedures. SPME technology is simple, fast, and environmentally friendly. However, it also has some disadvantages, such as the fiber being easily broken, and how multiple uses may exhibit the carryover phenomenon. To overcome these shortcomings, a novel micro solid phase extraction (μ-SPE) technique was developed. It packed a small amount of adsorbent with different formats, such as a sealed tea bag, in-syringe, and in a polypropylene membrane envelope, in an attempt to shrink the traditional SPE to a smaller size. They provide easy procedure and short sample preparation time. In the study of Chmangui et al. [113], the adsorbent MIP was enclosed in a cone-shaped polypropylene membrane, preparing the porous membrane-protected MIP-μ-SPE to assess trace levels of AFB1 and B2 in non-dairy beverages. After 15 min of extraction, the EF of this method was 33.3, and analytical recoveries were in the range of 91–104%. In order to improve the extraction efficiency, a combined procedure was developed for the preparation of mycotoxins. Jayasinghe et al. [98] developed a UAE-MIP-μ-SPE method to extract AFs in fish feed. The MIP adsorbents were protected in a porous membrane. Under the optimized conditions, RSDs below 20%, an EF of 33.3, and analytical recoveries in the range of 80–100% were obtained. In addition, the devices can be reused 20 times (20 loading/eluting cycles) without losing its accuracy, which provided a practical advantage over commercially available disposable SPE cartridges.

#### 2.2.6. Summary and Recommendations

SPE, as the most commonly used pretreatment method for the extraction of mycotoxins, has the advantages of multiple types of stationary phases, high extraction efficiency, and automation. The development of a variety of emerging materials, such as MIPs, GO, and HCPs, as well as new methods like MSPE, will facilitate rapid, sensitive and green pretreatment of mycotoxins. SPME technology simplifies the operation process, and reduces the consumption of organic reagents and adsorbents, which overcomes the shortcomings of traditional SPE methods where the pores of the filler are easily blocked. However, the short life of the adsorbent material, low selectivity, and cross-memory of SPME limit the extensive use of this method. New techniques (such as D-μ-SPE and μ-SPE) have broad application prospects in the preparation of mycotoxins. In the next few years, the development of highly selective and environmentally friendly adsorbents will continue to be an important research area based on SPE technology. In addition, the miniaturization of equipment, the simplicity of operation, the integration and automation of pretreatment, and detection technologies will also become the focus of the study.

### 2.3. LLE-Based Approaches

#### 2.3.1. Salting-Out Assisted Liquid-Liquid Extraction (SALLE)

At present, the pre-treatment method is mainly SPE, but this method is not suitable for high-throughput detection due to its high-test cost and relatively complicated operation. Salting-out assisted liquid-liquid extraction (SALLE) facilitates better separation of the extraction solvent from the mixture by adding appropriate inorganic salts to the mixture of aqueous samples and water-miscible organic solvents. The method has the advantages of simple operation, low organic reagent consumption, and high extraction efficiency. Hamed et al. [77] established a SALLE method to extract *Fusarium* toxins in functional vegetable milks with no further clean-up. ACN containing 5% FA was added to the solution containing the sample and potassium dihydrogen phosphate, and then MgSO_4_, NaCl, sodium citrate, and disodium hydrogen citrate were added to promote the extraction of the target. Under the optimized conditions, recoveries above 80% and RSD lower than 12% were obtained. This shows that SALLE is a simple and effective sample treatment strategy for samples that do not need cleaning.

#### 2.3.2. Single-Drop Microextraction (SDME)

Single-drop-liquid-liquid microextraction (SD-LLLME), as a simple, rapid, low-cost, and minimal residual sample pretreatment method, has been used to extract a variety of samples. In the SD-LLLME process, the analyte is first extracted from the aqueous sample phase into the upper organic solvent layer, and then back-extracted into the droplets of the acceptor phase suspended in the organic phase. In this way, highly polar compounds (such as sugar) will remain in the sample phase, and nonpolar compounds will remain in the organic phase [131]. Therefore, the MEs from the sample can be significantly reduced, and the effective extraction, purification, and enrichment of the target can be realized. Li et al. [87] applied a simple SD-LLLME method to pretreat patulin in apple juice. Compared with other methods (which usually need complicated pretreatment, including extraction, purification, evaporation, and redissolution), the whole sample pretreatment of SD-LLLME only consumed 1.5 mL ethyl acetate and 20 min using a single extraction. This method could greatly lower the interferences from sugar-rich matrix, which realized the simple, rapid, and green extraction of the targets.

#### 2.3.3. Dispersive Liquid-Liquid Microextraction (DLLME)

In recent years, liquid phase microextraction (LPME) technology has become popular because of its low consumption of organic reagents and high preconcentration coefficient. Among them, dispersed liquid-liquid microextraction (DLLME) as a simple, rapid, low-cost, and high-recovery sample pretreatment method is getting more and more applications. DLLME systems consist of three parts: extraction solvent, dispersion solvent, and sample solution. Figure 1C shows the common DLLME process. Compared with other methods, the sufficient contact area between the target and the fine droplets accelerate the extraction process and significantly shortens the analysis time. Salim et al. [26] developed a DLLME method to extract multi-mycotoxin in rice bran. In the DLLME process, MeOH/H_2_O (80:20, *v*/*v*) and chloroform were used as the dispersive solvent and extraction solvent, respectively. After optimization, recoveries ranging from 70.2 to 99.4% with an RSD below 1.28% were obtained.

However, in the traditional DLLME process, the mixture of dispersant and extraction reagent is quickly injected into the sample solution manually through a micro-syringe. This operation makes the size of the generated droplets uncontrollable, resulting in insufficient surface contact between the extraction solvent and the water phase. In order to maximize the extraction efficiency, several methods of dispersing solvents by kinetic energy have been reported in recent years, such as air-assisted DLLME (AA-DLLME) [27], ultrasonic-assisted DLLME (UA-DLLME) [75], and vortex-assisted DLLME (VA-DLLME) [28]. Karami-Osboo [27] established an AA-DLLME method to purify and enrich AFB1, B2, G1, and G2 from rice samples. Air-assisted dispersion was used to replace the dispersion solvent in the conventional DLLME system to form a cloudy solution. Under the optimized conditions, chloroform had the highest performance in the extraction of AFs. Compared with IAC clean-up (recovery: 76–89%), AA-DLLME obtained a higher recovery of 90–106%.

Inkjet printing is an important industrial technology, which can accurately control the droplet volume from nanoliter to picoliter level. This is one of the most promising and efficient droplet generation methods at present [132]. Based on these characteristics, the combination of inkjet printing technology and DLLME technology can not only improve the pretreatment speed and reduce the consumption of organic reagents, but also realize more accurate sample injection and automation. Sun et al. [47] developed an inkjet-based DLLME method to extract AFs from wheat. A drop-on-demand jetting device was used in this process, and the extraction solvent (only 10 μL) was injected into the sample solution as ultrafine droplets (about 20 μm in diameter) at high frequency to form a cloudy liquid. After optimization, satisfactory recoveries between 83.2 and 93.0% with RSD below 4.6% were obtained for all compounds. This convenient and reliable method is environmental-friendly, representing a new development direction of traditional DLLME technology.

Another improved strategy to increase the extraction efficiency of DLLME and realize greener analysis is the use of low toxicity or nontoxic extraction solvents, such as ILs [48], DESs [75], and low-density solvent (LDS) [28]. Somsubsin et al. [28] developed a VA-DLLME using nontoxic LDS as extraction solvent to pretreat AFB1, B2, G1 and G2 in rice samples. In this process, vortex stirring was used for dispersion without adding dispersion solvent. In addition, adding salt (Na_2_SO_4_) as a demulsifier for rapid phase separation can avoid centrifugation. Four different extraction solvents with lower density than water including 1-dodecanol, 1-undecanol, 1-octanol, and toluene were studied and compared. Consequently, 1-octanol/toluene (1:1, *v*/*v*) was selected as the extraction solvent. Under the optimal conditions, high EFs (42–132), good precisions (RSDs < 6.2%), and satisfactory recoveries (70–104%) were obtained. In addition, this method was comparable to the IAC method, but the current method is faster and more cost-effective than IAC.

### 2.4. QuEChERS

Nowadays, QuEChERS is the preferred clean-up approach because it is quick, easy, cheap, effective, rugged, safe, and suitable for multi-analyte determination. QuEChERS method was first developed in 2003 by Anastassiades et al. to extract pesticide residues from fruits and vegetables [133]. The technology includes: (1) liquid-liquid distribution, in which ACN is usually used to extract the targets in a water environment. After shaking, anhydrous MgSO4 and NaCl are added to promote the water partition from the organic phase and its dehydration; and (2) a dispersive solid phase extraction (d-SPE) method with primary secondary amine (PSA) or C_18_ sorbent is often used to purify the extract. In this extraction process, selecting the appropriate extraction solvent and d-SPE sorbent are the key factors affecting the extraction results. He et al. [35] used a QuEChERS method to extract DON and ZEA in cereals. After optimization, MeOH/ACN (8:2, *v*/*v*) was chosen as the extraction solvent, vortex oscillation showed higher recoveries (79.25–101.72%) than ultrasonication (78.90–88.30%) and hand agitation (72.02–80.91%), PSA (300 mg) was chosen as the best sorbent, and 60 s was selected as the clarification time. Under the optimized conditions, satisfactory recoveries (83.55–106.93%) and low coefficient of variation (CV) values (<5.19%) were achieved.

However, traditional QuEChERS method using PSA or C_18_ as adsorbents are usually inefficient to overcome MEs in samples with a high lipid composition. To solve this problem, modified QuEChERS methods are developed. Alcántara-Durán et al. [56] established a QuEChERS method with EMR-lipids as the adsorbent for the pretreatment of 16 multiclass mycotoxins in edible nuts. Compared with traditional mixtures of PSA and C_18_ d-SPE sorbents, EMR-lipid achieved negligible MEs for all mycotoxins studied in peanuts, pistachios and almonds, and satisfactory recoveries ranging from 75 to 98% was obtained.

### 2.5. Matrix Solid Phase Dispersion (MSPD)

In MSPD, a small amount of samples are evenly mixed with solid support, and then the obtained powder is eluted with solvent, which realizes the simultaneous separation, extraction and purification of the target in solid and semi-solid samples. The procedure of MSPD is shown in Figure 1D. Because a small amount of solid support and solvents are used, the cost of analysis is reduced, and it is an environmentally friendly sample pretreatment method. In addition, the use of the dispersant of MSPD greatly increases the contact area with the target molecule and realizes rapid dissolution. de Oliveira et al. [43] developed a MSPD method to extract FB1 and B2 in different types of maize. After optimization, silica gel was selected as the dispersant, the extraction solution only contained 30% of organic solvent, and a lower elution volume (6 mL) was used. Trueness (recoveries: 86–106%) and precision (RSD < 19%) was achieved. To further improve extraction efficiency, vortex-assisted MSPD was used by Massarolo et al. [50] for the extraction of AFG2, G1, B2, and B1 in cornmeal. Using 25 mg C_18_ as solid support, 10 mL of ACN/MeOH (1:1, *v*/*v*) as elution solvent, and vortexed for 3 min, the best recoveries ranging from 85.7 to 114.8% were obtained, which indicated that this method was suitable for coarse and medium grind cornmeal.

### 2.6. Dilute-And-Shoot

However, some of the above methods will have co-extractive problems, resulting in strong matrix interference. The “dilute-and-shoot” method can reduce the MEs as much as possible, and has the advantages of simplicity, less analyte loss and high sample throughput. In the work of Castilla-Fernández et al. [80], after SLE was used to extract mycotoxins from walnuts, six different clean-up approaches specially designed for fatty matrices, including SPE with PRiME HLB or AFFINIMIP cartridges, dSPE with Z-sep+, C18, EMR-Lipid, or PSA were tested to reduce MEs. However, none of these methods can effectively reduce MEs (40–83%). As an alternative, the dilute-and-shoot approach was investigated. It is exciting that the MEs (<20%) obtained by using SLE extract with 1:100 dilution was weak or negligible.

### 2.7. Hybridization of Different Sample Pretreatment Methods

To reduce matrix interference as much as possible and achieve higher recovery and sensitivity, a combination of different sample pretreatment methods (combining the advantages of each method) is a good choice. Nouri and Sereshti [54] developed a fast method based on in-syringe SPE combined with DLLME to prepare Afs in soybean. PU/GO electro-spun nanofibers coated on a metal net sheet was used as the adsorbent of SPE. After optimization the parameters of SPE and DLLME, satisfactory relative recoveries (76–101%) were achieved. In addition, LLE or SLE has a limited ability to remove matrices, which usually requires an SPE clean-up program [62,97]. This section is divided by subheadings. It should provide a concise and precise description of the experimental results, their interpretation, as well as the experimental conclusions that can be drawn.

### 2.8. Recommendations

For different substrates and targets, the adaptability of sample pretreatment methods should be carefully considered to obtain a satisfactory recovery and meet the requirements of determination methods. In this paper, various pretreatment methods of mycotoxins in different substrates are systematically summarized and compared. The advantages and disadvantages of different pretreatment technologies are summarized in Table 3. In the past five years, SPE and QuEChERS are still the most commonly used pretreatment methods of mycotoxins because they have been commercialized. In order to pursue higher efficiency, emerging materials are a research hotspot, which are often used as fillers of SPE or adsorbents of MSPE to realize rapid and green enrichment and purification of mycotoxins. However, new materials may face complicated preparation processes, high energy and time consumption, high cost, and environmental pollution problems, which need more in-depth discussion by researchers in this field to realize the effective application of these materials. As for microextraction, its application in mycotoxins is limited at present, but it can obviously reduce the amounts of organic reagents and materials, which conforms to the guiding principles of green chemistry, and this is undoubtedly a major development direction in the future.

AFs are the most widely studied mycotoxin in recent years. Almost all the sample pretreatment methods introduced in this paper are suitable for the pretreatment of AFs in different matrices. Notably, for solid substrates, such as rice and grain, the traditional SLE program is a pre-extraction strategy, while for liquid substrates, such as oil and milk, the traditional LLE program is a pre-extraction strategy, followed by SPE, MSPE, or other methods to purify and enrich toxins. As far as the other mycotoxins, like OTA and ZEA, different extraction and clean-up techniques, such as IAC, SPE with various alternative stationary phases, DLLME, SPME, IT-SPME, QuEChERS, on-line SPE, and so on, can be chosen based on time, cost, sample throughput, and automation requirements. Some special toxins, such as FB1, have a large molecular weight and multiple interaction sites on the molecular chain, which makes them interact more with the matrix, thus increasing the difficulty of extraction. In the choice of organic solvent, compared with ACN, MeOH is more polar and acidic, and the higher the proportion used in the extraction process of FB1, the higher the recovery.

## 3. Conclusions

Mycotoxins have attracted wide attention because of their toxic effects on the human body and their harmful effects on crops. Recently, different sample preparation techniques have been widely used to extract and concentrate mycotoxins. IAC and QuEChERS are two commonly used methods to prepare mycotoxins. However, they have some disadvantages, such as difficulty in automation or a high internal cost. SPE has the advantages of multiple types of stationary phases, high extraction efficiency, and automation. Newly developed microextraction technologies, such as SPME, D-μ-SPE, and μ-SPE, can extract the mycotoxins conveniently and efficiently, reduce the consumption of organic reagents and adsorbents, and conform to the concept of modern green chemistry. The appearance of MSPE and DLLME technology simplifies the process of extraction, purification and enrichment of mycotoxins in complex matrices. In addition, new green pretreatment reagents such as ILs, DESs and LDS, and cutting-edge materials like MIPs, MWCNTs, COFs, and MOFs have been developed for the pretreatment of mycotoxins. Pretreatment methods, reducing solvent consumption, sampling volume, analysis time, operation cost, realizing automation, and developing new materials are research hotspots in this area.

## Figures and Tables

**Figure 1 toxins-15-00215-f001:**
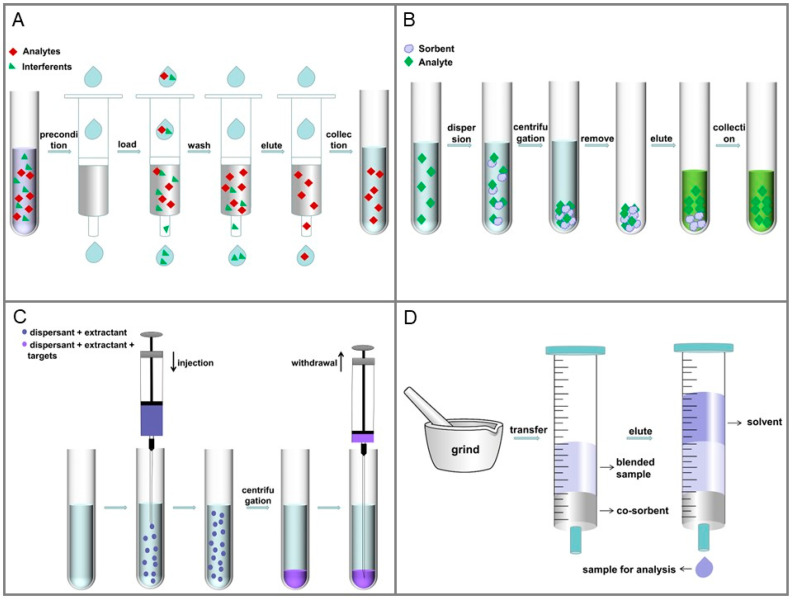
The procedure of different pretreatment methods. (**A**) Schematic diagram of SPE. (**B**) The main steps of D-μ-SPE. (**C**) Diagram of the extraction process of DLLME. (**D**) Schematic representation of the MSPD procedure.

**Figure 2 toxins-15-00215-f002:**
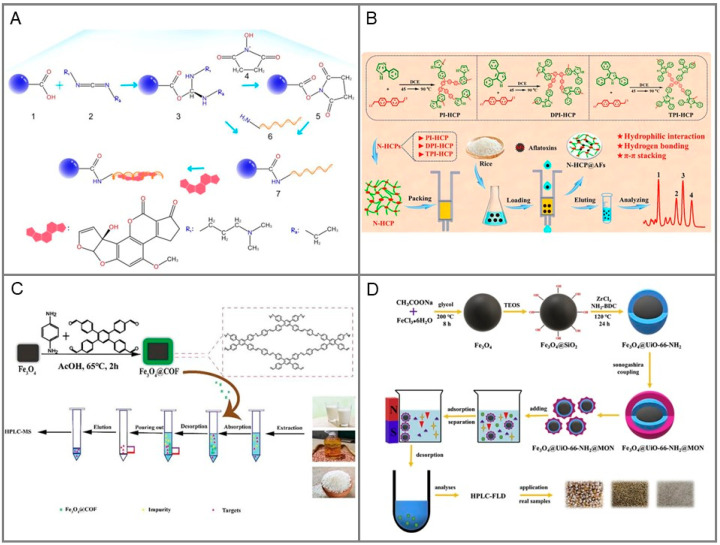
Preparation methods of different new materials. (**A**) Synthesis of M1-aptamer-modified microspheres [70]. (**B**) The preparation of N-heterocyclic HCPs and their application for extraction of AFs [18]. (**C**) Construction of Fe_3_O_4_@COF(TFPB-PPD) [23]. (**D**) Schematic illustration of the synthesis and application of Fe_3_O_4_@UiO-66-NH_2_@MON [24].

**Table 1 toxins-15-00215-t001:** SPE conditions for the pretreatment of mycotoxins in different food matrices.

Matrix	Analytes	Cartridge/Sorbent	Precondition	Wash	Elution	Recovery	RSD	Ref.
rice and fragrant rice	AFB1, B2, G1, G2	PI-HCP column	MeOH, acetone and H_2_O	/	acetone	82.2–113%	<9.1%	[18]
rice and noodle products	AFB1, B2, G1, G2	HLB or PRiME HLB column (collect filtrate directly)	n.d.	MeOH/H_2_O (20/80, *v*/*v*)	MeOH	80.5–106.6%	2.4–7.2%	[19]
rice and wheat	AFB1, B2, G1, G2	graphene	n.d.	/	ACN:H_2_O:MeOH: acetic acid(59.4:9.9:29.7:1 *v*/*v*)	70.61–113.30%	<6.13%	[20]
cereal products	HT-2 and T-2	IAC	pure H_2_O	n.d.	ethanol	78.6–98.6%	1.2–6.8%	[30]
cereals	AFB1, B2, G1, G2	MIL(Al)-53-DES@MIPs	n.d.	distilled H_2_O	ACN-H_2_O (9:1, *v*/*v*)	95.3–98.5%	1.3–4.4%	[31]
corn and peanut	AFB1	Sil@HIm-Im column	n.d.	ACN and H_2_O	MeOH/acetic acid (2.0% vol.)	80.0–103.3%	2.37–6.58%	[36]
corn and corn products	FB1 and FB2	SAX cartridge	MeOH and MeOH/H_2_O (75/25, *v*/*v*)	MeOH/H_2_O (75/25, *v*/*v*) and MeOH	1% formic acid in MeOH	79.4–98%	3.5–55.7%	[38]
foodstuffs	AOH and AME	MIPs column	MeOH and phosphate buffer (50 mmol/L, pH = 8.2)	ACN/water (5:95, *v*/*v*), ACN/water (15:85, *v*/*v*)	1% TFA in MeOH	92.5–106.2%	<20%	[60]
peanut oils	AFB1	Carb/PSA column	n.d.	normal hexane	MeOH-dichloromethane (2/8, *v*/*v*)	87.7–105.1%	2.2–7.9%	[62]
edible oils	AFB1, B2, G1, G2	HAS column	acetone/H_2_O (8/2, *v*/*v*) and n-hexane	iso-propanol	acetone/H_2_O (8/2, *v*/*v*)	85–100%	<11%	[63]
milk and dairy products	AFM1	IAC	n.d.	distilled H_2_O	MeOH	85.2–107.0%	≤7 %	[69]
milk	AFM1	AFM1-aptamer modified microspheres	n.d.	5% MeOH-H_2_O	10 mM Mg^2+^ and ACN-MeOH-H_2_O (*v*/*v*, 2:1:1)	85.3–109.9%	2.6–6.7%	[70]
milk	9 mycotoxins	rGO/Au column	/	MeOH/H_2_O (5/95, *v*/*v*)	MeOH/ACN/formic acid (50/49/1, *v*/*v*/*v*)	70.2–111.2%	2.0–14.9%	[71]
nuts	AFB1, B2, G1, G2	AC-B column	n.d.	n.d.	n.d.	89.3–96.1%	0.3–7.0 %	[81]
fruits and vegetables	7 mycotoxins	HLB SPE cartridge	n.d.	1% formic acid in H_2_O	MeOH	81.1–116%	3–6.2%	[85]
feed	FB1 and FB2	IAC	n.d.	0.01 M PBS	MeOH and distilled, deionized	FB1: 75.1–109%; FB2: 96–115.2%	1.0–16.7%	[96]
animal feed and food	11 mycotoxins	EZ-Pop NP column	acetone	/	ACN	70–120%	<20%	[97]
beer, red wine, corn, and Turkish coffee	OTA	IAC	n.d.	PBS (pH: 7.4)	MeOH/HAC (98:2, *v*/*v*)	104.34–107.33%	0.21–1.31%	[104]
dark tea	AFB1, B2, G1, G2	MFC-IAC	n.d.	n.d.	n.d.	77.5–93%	2.2–11%	[108]
edible and medicinal herbs	6 AFs and 6 ZEAs	IAC	n.d.	PBS and 5 mL H_2_O	MeOH	64.7–112.1%	<13.7%	[115]
human urine	ZEA, α-ZEL, β-ZEL, α-ZAL, β-ZAL, ZAN	96-well μElution	MeOH and H_2_O	H_2_O and 50% MeOH	H_2_O	87.9–100%	<7%	[120]
pig hair	FB1	SAX clean-up column	MeOH and MeOH:H_2_O (3:1, *v*/*v*)	MeOH:H_2_O (3:1, *v*/*v*), MeOH	MeOH:acetic acid 0.5%	70–106%	1.0–5.0%	[121]

n.d.: Not discovered; /: This step is not required. MeOH: methanol; ACN: acetonitrile; PI-HCP: phenyl-imidazole hyper-crosslinked polymer; HAS: humic acid-bonded silica; PBS: phosphate buffered saline; MFC: multifunctional purification column.

**Table 2 toxins-15-00215-t002:** SPME conditions for the pretreatment of mycotoxins in different food matrices.

Matrix	Analytes	Sorbent	Volume/mg	Adsorption Time/min	Elution	Recovery	RSD	Ref.
rice and sorghum	AFB1, B2, G1, G2	MHCP-TPE	30	10	ACN	81.9–117.0%	<8.0%	[21]
rice and maize	AFB1, B2, G1, G2	M-OP10-DCX	25	10	ACN	82.8–115%	<8%	[22]
milk, edible oil and rice	AFB1, B2, G1, G2	M-COF	2	2	ACN	76.4–112.5%	<15%	[23]
corn, rice and millet	AFB1, B2, G1, G2	Fe_3_O_4_@UiO-66-NH_2_@MON	10	10	ACN	87.3–101.8%	2.2–3.0%	[24]
rice	AFs	Fe_3_O_4_/zeolite nanocomposite	50	1	MeOH	80–104%	1.8–7.2%	[29]
cereals	AFB1, B2, G1, G2	Fe_3_O_4_@POSS@PIL-PSt	80	5	ACN	87–120%	3.2–11.2%	[33]
maize	AFs, OTs and enniatins	Fe_3_O_4_/COF-TpBD	5	0.5	ACN/H_2_O/ acetic acid (85:10:5)	73.8–105.3%	<8.5%	[41]
maize	ZEA and its derivatives	MWCNT-MNPs	20	3	acetone containing 0.5% formic acid	75.8–104.1%	≤14%	[42]
maize, wheat, watermelon andmelon	AFB1, B2, G1, G2, OTA, OTB, T-2, HT-2 and DAS	MIL-101(Cr)@Fe_3_O_4_	25	4	acetone containing 1% formic acid	83.5–108.5%	1.6–10.4%	[45]
cornmeal	ZEN	immunomagnetic chitosan	100	1	MeOH	91.7–104.3%	2.9%	[51]
foodstuffs	AFB1, B2, G1, G2, AFM1, and AFM2	AF-mAb/CTS/Fe_3_O_4_	0.3 mL	0.5	MeOH	63–118%	≤ 16.3%	[61]
edible vegetable oils	AFB1, B2, G1, G2	PDA@Fe_3_O_4_-MWCNTs	50	10	ACN/water/acetic acid (84:15:1)	70.15–89.25%	≤6.4%	[64]
vegetable oil	FB1, ZON and OTA	Fe_3_O_4_@nSiO_2_@mSiO_2_	5	10	ACN/MeOH (1:1) containing 1% formic acid	85.0–94.7%	3.1–5.3%	[65]
vegetable oils	AFB1, B2, G1, G2	bare Fe_3_O_4_ nanoparticles	10	10	n-hexane	82.6–106.2%	≤9.8%	[66]
milk and yogurt	6 mycotoxins	core-shell poly(dopamine)	60	0.5	MeOH	70–120%	≤16%	[72]
nuts	AFB1, B2, G1, G2	Fe_3_O_4_@SiO_2_@TiO_2_-APTMS-CPA	10	2	MeOH	87.7–97.5 %	<7.1%	[82]
*Salviae miltiorrhiza Radix et Rhizoma* (Danshen)	ZEA, T-2, HT-2, NEO, DAS	Fe_3_O_4_/MWCNTs	20	n.d.	acetone containing 0.5% formic acid	73.7–91.9%	2.1–13.3%	[116]

n.d.: Not discovered. MHCP-TPE: magnetic hyper-crosslinked polymer; BD: benzidine; TP: 1,3,5-triformylphloroglucinol; PIL: polymeric ion liquid; PSt: polymeric styrene; POSS: polyhedral oligomeric silsesquioxanes; MON: microporous organic network; CTS: chitosan; CPA: 1,8-bis (3-chloropropoxy) anthracene-9,10-dione; APTM: 3-aminopropyltrimethoxysilane.

**Table 3 toxins-15-00215-t003:** The advantages and disadvantages of different pretreatment technologies.

Pretreatment Methods		Advantages	Disadvantages
SPE-based approaches	SPE	Effective preconcentration; High recovery, efficiency, and selectivity; Easy to realize automation.	Easily clogged the filler pores; Need a lot of organic reagents, causing environmental pollution.
	MSPE	Fast and simple; Environment friendly; Reusable.	Complex material preparation; High cost.
	SPME	Less consumption of organic reagents; Simpler to operate and implement; Easily eluted and high recovery.	Fibers easily degraded (multiple uses); Carryover phenomenon occurs between extractions; Limited adsorption capacity.
	D-μ-SPE	Miniaturized extraction; Simple to operate; No solvent and micro- or nano-sorbents.	High aggregation tendency (nano-adsorbents); Need external energy.
	μ-SPE	Avoid fiber degradation and carryover phenomenon.	The sorbent floats or sticks to the wall.
LLE-based approaches	SALLE	Simple equipment and convenient operation; Wide application.	Mixed with a lot of neutral salt.
	SDME	Simplicity and inexpensive; Less solvent consumption and higher enrichment factor; Compatibility with GC and LC systems.	Instability of the droplet; Limited types of extractants; Long extraction time.
	DLLME	Simplicity of operation; Low cost; High recovery, efficiency and enrichment factor.	Nonselective extraction; Difficulty of automation; Need a third component.
QuEChERS		Quick, easy, cheap, effective, rugged and safe.	Matrix dependencies; Need a lot of organic reagents; Difficulty in automation.
MSPD		Simple structure and wide application; Short extraction time; Low solvent consumption.	Limited adsorption capacity
Dilute-and-shoot		Reduce matrix effects; Simplicity of operation.	Low recovery and accuracy

## Data Availability

No new data were created or analyzed in this study. Data sharing is not applicable to this article.

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
