# Peer review of "Recent Insights into Sample Pretreatment Methods for Mycotoxins in Different Food Matrices: A Critical Review on Novel Materials"

_toxins, 2023, doi:10.3390/toxins15030215_

Round 1

Reviewer 1 Report

Title: Recent Insights into Pretreatment Methods for Mycotoxins in Different Food Samples: A Critical Review on Novel Materials

Journal: Toxins

Reference: toxins-2261295

The authors present a comprehensive summary of pretreatment methods for mycotoxin determination including solid-phase extraction (SPE), liquid-liquid extraction (LLE), matrix solid phase dispersion (MSPD), QuEChERS, among others. Also, novel materials and cutting-edge technologies are systematically summarized. In general, this work provides plenty of information on the topic and it is suitable for this journal. However, some minor problems, in my opinion, should be solved before consideration for publication.

 L21. Mycotoxin is a low-molecular-weight (about 700)… 700 Da?

L22. Contamination instead of pollution

L37. Fusarium should be in italic, similarly hereinafter.

L39. altenuene (ALT)

L65-67. The meaning of the sentence is not clear.

L78. mycotoxins absorbed in the sample… absorbed or deposited?

L89. corn meal

L127. 1% FA… 1% fusaric acid?

L148. Please provide the meaning of all abbreviations presented in Tables 1 and 2!

L243-245. The meaning of the sentence needs to be clarified.

L261. (3 min)

L295-6. [2-phenylimidazole (PI), 4,5-diphenylimidazole (DPI) and 2,4,5-triphenylimidazole (TPI)]

L299-302. The meaning of the sentence is not clear.

L308. Avoid the usage of the words “obvious” or “obviously” (throughout the manuscript).

L359. (MMOFs)

L407-409. Please revise this sentence.

L432. The “program” is shown in Figure 1B. Which program?

L450-2. The same as L295-6.

L582. was chosen…

L608. C18

L616. Castilla-Fernández…

L634. More emphasis on finding and its implication may be mentioned in the conclusion section. For instance, what about the complex preparation processes, high energy/time consumption, and the environmental hazards of certain proposed methodologies to prepare the novel materials? These aspects inevitably complicate its large-scale application.

Reviewer 2 Report

Title: Recent Insights into Pretreatment Methods for Mycotoxins in Different Food Samples: A Critical Review on Novel Materials

Review Report: The topic is of interest for people working with food safety especially mycotoxins. The overall content is effective only if it will elaborate with specifically with particular mycotoxin, for example aflatoxins what are effective ways for the extraction of aflatoxins from different matrixes like spices, grains, oils and feed etc, similarly ochratoxin A, how pretreatments are more effective for OTA in different food stuff? I think this would create more depths in your review article.

In Tables it would be more efficient if we arrange same matrix for different toxins extractions, it will create more clearness in review article.

Overall the content is manuscript is presented efficiently, and it is useful for scientists, researchers and food technologists.  

Reviewer 3 Report

Authors have presented an overview of the current different methods used to extract mycotoxins from different food samples. The review goes over a dozen technologies/methods that can be used to potentially improve mycotoxin extractions from difficult or complex food matrixes. While showcasing numerous examples of methodology and papers extracting mycotoxins in these matrixes, the paper lacks direction for someone to go after reading. If I was trying to gain insight on how to extract fumonisin from a difficult matrix, this paper would talk about all the different things I could try without suggesting one as an avenue to pursue.

With the focus of paper being preparing samples for mycotoxin extraction, the conclusion does very little to summarize how the technologies listed are directly related to mycotoxins. If this was just a review of the capabilities of these current technologies, then this review achieved that. This section could even replace the word “mycotoxins” with “sample” and in most cases lose no meaning. The abstract talks about suggesting prospects for these different methods presumably with mycotoxins in mind. This aspect needs to be added as it is not present in my opinion. I encourage the authors to talk about the five broad pretreatment methods listed in Table 3 with their use in mycotoxins. Regarding extracting mycotoxins in food, industry is usually interested in speed, cost efficacy, and simple/robust methods of identifying mycotoxins. This would be an interesting area to expand and is not well defied in comparing the technologies with mycotoxins in mind. For example, multiple pretreatments are stated to reduce the consumption of reagents used in the extraction but do not indicate how much even in a qualitative way for comparison.

Another item that should be addressed is the use of the word “Pretreatment” in the title and paper. changed. The idea of pretreatment is a little vague and can allude to multiple things with mycotoxins.  For instance, I thought initially based on the title that this article was about reducing mycotoxin levels in contaminated food. I recommend changing the wording to sample pretreatment in the title and as a keyword.

Minor issues with Manuscript

Line 21 presume 700 means 700 Daltons. Please add units.

Sentence in line 2 what do you mean by especially apples? Do you mean that apples are particularly targeted in terms of fruit in mycotoxins?

A citation for the sentence that begins on this line 28 is needed.

Cut back on number of abbreviations. There are quite a few cases of where an abbreviation is only used a single time i.e., SWE and PHWE. Abbreviations should be used more than once to justify their use otherwise it starts to impact the readability and clarity of the paper.

Please check figures that they meet the journal standards. Several figures seem to be blurry especially when those that have small font size as the word are hard as clear as they should be.

The tables should be formatted differently as they are hard to read and visually tell the rows apart, especially where multiple rows correspond to a single entry. Simple fix would be to outline each entry and use the lines to separate the data.
